# Thermal Characteristics and Kinetics of the Thermal Degradation of Sugar Beet Waste Leaves and Pulp in Relation to Chemical Composition

**DOI:** 10.3390/foods14020307

**Published:** 2025-01-17

**Authors:** Sanja Ostojić, Darko Micić, Josipa Dukić, Iva Sabljak, Ayça Akyüz, Seda Ersus, Anet Režek Jambrak

**Affiliations:** 1Institute of General and Physical Chemistry, Studentski Trg 12/V, 11000 Belgrade, Serbia; micic83@gmail.com; 2Department of Food Engineering, Faculty of Food Technology and Biotechnology, University of Zagreb, Pierotti Street 6, 10000 Zagreb, Croatia; jdukic@pbf.hr (J.D.); anet.rezek.jambrak@pbf.unizg.hr (A.R.J.); 3Eurofins Croatiakontrola d.o.o., Karlovačka Cesta 4L, 10000 Zagreb, Croatia; iva.sabljak@ftcee.eurofins.com; 4Department of Food Engineering, Ege University, İzmir 35040, Bornova, Turkeysedaersus@gmail.com (S.E.)

**Keywords:** sugar beet leaves, by-products, thermal analysis, DSC, TGA, kinetics, activation energy, enthalpy of transition, thermal degradation

## Abstract

Thermal characteristics of dried sugar beet pulp, leaves and leaf fractions obtained after extraction: fibrous leaf pulp and fibre rich leaf fraction, were investigated by differential scanning calorimetry and thermogravimetry. The sugar beet samples showed a similar thermal behaviour associated with a similar composition. Two endotherms are found on the differential scanning calorimetry curves. First one in the temperature range 31–153 °C and the second from 150–160 °C. Thermal degradation kinetics was studied by thermogravimetric analysis. Four degradation stages were observed within the temperature range 25–700 °C. The kinetic parameters of the degradation, obtained by Ortega and Friedman non-isothermal isoconversional methods did not significantly differ between models: Ea-activation energy at a conversion degree 0.1–0.9 ranged 50–200 kJ/mol; lnA- the natural logarithm of the pre-exponential factor 8–48; kp_1_-thermal degradation rate constant at a conversion extent of 0.5 ranged of 0.19–2.55 min^−1^. Constant rate of degradation is highest for the sugar beet leaves kp_1_ (2.58–2.55 min^−1^), and kp_2_ (70.1–70.4 min^−1^). The results obtained are valuable for sugar beet leaf industrial processing. A positive environmental impact is achieved by transforming the waste into high-value food additives.

## 1. Introduction

Sugar beet (*Beta vulgaris* L. *vulgaris*) is a biennial plant species of the *Amaranthaceae* family and is one of the four currently cultivated subspecies of beet and the only one used for sugar production [1]. Sugar beet cultivation is widespread worldwide, but sugar beet generally thrives best in temperate climates between 30 and 60 ° north latitude [2]. In addition to sugar production, sugar beet is also important as a crop grown in rotation with other crops [3]. Today, sugar beet is the second most widely cultivated crop for sugar production, with the first being, of course, sugar cane, which accounts for around 80% of total global sugar production [4]. Most sugar beet is grown in European countries, i.e., European Union (EU) countries, Russia and Ukraine, and according to the European Commission (EC), EU countries account for around 50% of global production of this valuable sugar source.

In general, food production generates certain by-products and waste. For example, the extraction of sugar from sugar beet also produces various by-products and waste at certain stages of the process. Beet pulp, a by-product of extracting sugar from sugar beet pulp, is usually pressed and used as animal feed. Recently, with the development of biotechnology, advanced methods of treating and fermenting this by-product have been considered. Reducing food waste not only benefits the environment but also contributes to various social agendas by reducing greenhouse gas emissions, enhancing ecosystem services, and enforcing sustainable resource management policies. Food waste accounts for 8–10% of global greenhouse gas emissions and ranks as the third-largest emitter globally, following China and the United States [5]. In order to use resources efficiently and enhance food security and sustainability, food waste and by-products must be reduced and properly valorized. Plant-based food production generates various by-products, which are generally rich in nutrients and bioactive compounds.

In order to achieve Sustainable Development Goals (SDGs), the food industry and other industries are trying to minimize the final amount of accumulated waste and by-products. This is being pursued through increased recycling efforts and repurposing these materials for alternative applications. In addition to the accumulation of a large amount of waste and by-products, sugar production also has a significant impact on the environmental factors. Namely, a life cycle assessment study found that the production of 1 ton of white consume sugar generates up to 1540 kg of CO_2_ equivalents and consumes up to 12.6 m^3^ of water [6]. Emerging technologies have been effectively employed to extract these valuable compounds with health benefits [5]. Therefore, new biotechnological processes aim to utilise this rich carbohydrate content as a potential source for the production of biofuels, biopolymers and additives in the food, pharmaceutical and chemical industries [7]. Sugar beet pulp contains 60–70% carbohydrates in dry matter, of which almost 90% is composed of cellulose, hemicellulose and pectin [8]. The use of hydrolyzed sugar beet pulp is also considered a useful medium for the production of microbial proteins [9]. Until now, it was common practice to use the remaining sugar beet leaves (waste) in the form of animal feed or to compost them in order to return the nutrients “lost” during beet growth to the soil and thus prepare it for the next harvest. Recent research on the possibility of utilizing waste generated during the production and processing of root vegetables, including sugar beet leaves, pointed to the numerous valuable components they contain, such as proteins, fibers and compounds with antioxidant activity [10,11,12]. In addition to their favorable protein composition, sugar beet leaves are characterized by a high content of structural carbohydrates in dry matter due to the biology of plant cells, which are an essential component of cell walls. Most of the structural carbohydrates are cellulose, hemicellulose and pectin, while lignin is present in small amounts [13]. Therefore, sugar beet leaves can be considered as a potential raw material for the sustainable production of bioethanol that complies with the rules of the circular economy [1]. In order to use sugar beet leaves as a potential raw material for the production of bioethanol, simple sugars, the building blocks of structural carbohydrates, must be obtained. Through the process of enzymatic hydrolysis, it is possible to depolymerize these carbohydrates into soluble sugars that can be converted into alcohol by yeasts and other fermentable microbial cultures. Aramrueang et al. [14] achieved the conversion of structural carbohydrates from sugar beet leaves into simpler sugars at a maximum of 82% after 72 h of enzymatic hydrolysis using three different enzyme preparations (CTec2, HTec2 and Pectinex Ultra SPL), demonstrating the possibility of using sugar beet leaves as a raw material for biorefinery when suitable enzyme preparations are used [14].

It is important to emphasize that sugar beet leaves, as a valuable food additive, have significant development potential in dietary supplements due to their high-quality biological components. Their use could improve overall human health and make sugar beet leaves an important ingredient in the production of functional foods [15]. In addition, the use of by-products of the food industry, such as sugar beet ingredients or apple residues [16], in various foods can not only increase their health benefits but also improve the technological properties of the produced foods. Moreover, the utilization of the bioactive compounds from sugar beet leaves (specific biological materials or by-products) can reduce waste generation and promote a more sustainable agricultural system by supporting a circular economy in which by-products and waste are efficiently reused [15].

The processing and use of food and food by-products in the food industry usually involve thermal treatment. Thermal properties of food systems are important in understanding the relationships between food properties and changes in food quality [17]. Heat treatment during food preparation can result in the degradation of certain components, such as polyphenols and polysaccharides [16] and proteins [18]. Consequently, studying the thermal characteristics of these components is crucial for their effective use as food ingredients. Monitoring thermal behaviour during heating helps identify the temperature limits for food processing and predicts the thermal stability and structural changes of food components. Understanding thermal properties is essential for designing optimal processing parameters to preserve the desired product quality. Thermal analysis provides valuable insights into the quality and stability of food products and by-products. It also aids in assessing shelf life, storage stability, and the impact of further processing at elevated temperatures [19].

The majority of studies dealing with the thermal analysis of plants and plant components in recent years have primarily focused on the thermal analysis of biomass, which represents a significant alternative renewable energy source. These studies are mostly concerned with the conversion of solid biomass, where it is necessary not only to understand the stages of the biomass combustion process but also to specifically comprehend the reaction kinetics and the release of volatiles [20].

On the other hand, there are studies that focus on the thermal characterization of waste and by-products from the fruit and vegetable industry, such as apple pulp [16], or the thermal characterization of dried extracts of medicinal plants [21]. The aim of these studies is to conduct thermal characterization to understand the thermostability of samples and to describe thermal properties related to the structural characteristics of the samples, which can significantly impact the quality of the final product as well as the technological processing of food. Thermoanalytical techniques have also been utilized to establish correlations between the chemical composition of plant samples and the results obtained through thermogravimetric analysis (TGA) [22]. These properties involve detecting transition temperatures of the sample, such as melting, glass transition, and thermal decomposition. Additionally, one of the crucial indicators of thermal behavior is the kinetic analysis of thermal decomposition. This knowledge can greatly contribute to forming the desired characteristics of the food sample.

A literature search revealed no studies investigating the thermal characteristics of sugar beet industry waste, specifically sugar beet leaves and pulp. The information obtained from the thermal characterization and the kinetics of thermal degradation of sugar beet leaves could be highly valuable for the development of functional food products derived from the sugar beet industry by-products/waste and also provide valuable insights for the sustainable production of bioethanol.

Therefore, it is of utmost importance to analyze the kinetics and thermal characteristics so that future potential non-thermal processing methods for the utilization of waste and by-products from the sugar industry can be theoretically optimized to ensure their reuse and reduce negative environmental impacts. The aim of the present study was to perform thermal characterization of dried sugar beet pulp, pellets, leaves, and various leaf fractions. Also, the study sought to determine the kinetic parameters of their thermal decomposition to gain deeper insights into the thermal degradation process.

## 2. Materials and Methods

### 2.1. Plant Material

All plant (*Beta vulgaris* L. *vulgaris*) materials (sugar beet leaves: SBL; fibrous leaf pulp: FLP; fibre-rich leaf fraction: FRLF, sugar beet pulp: SBPU and sugar beet pellets: SBPE) were provided by project partners from Turkey (Kayseri Şeker, Kocasinan Kayseri, Turkey and Ege University, Izmir, Turkey).

#### 2.1.1. Preparation of Dried Sugar Beet Leaves (SBLs)

The sugar beet leaves were collected and dried at room temperature until the dry matter reached at least 90%. After drying, the sugar beet leaves were ground (Retsch GM 200, Haan, Germany) and sieved to a maximum particle size of 500 µM. All experiments were conducted in triplicate, and quality control was ensured by consistently monitoring sample weights, drying times, and temperatures.

#### 2.1.2. Preparation of the Sugar Beet Leaf Pellets (SBPE)

Fresh sugar beet leaves were processed using a Homend juicer (Istanbul, Turkey) to separate the liquid (juice) and solid (leaf pellets) components. The extracted juice was set aside, while the remaining leaf pellets were dried in a tray dryer (Alveo, Konya, Turkey) at 55 °C for 48 h. All experiments were conducted in triplicate and quality control was ensured by consistently monitoring sample weights, drying times, and temperatures. Once dried, the pellets were stored in a refrigerator for future use.

#### 2.1.3. Preparation of the Sugar Beet Pulp (SBPU)

The sugar industry by-product, sugar beet pulp (SBPU), was prepared using the following process: sliced sugar beet roots were exposed to hot water (70–73 °C) at a pH of 5.5–6.0 for approximately one hour in a tower diffuser to extract the juice from the root cells; following diffusion, the wet sugar beet pulp was pressed and then dried in a tray dryer (Alveo, Konya, Turkey) at 55 °C, which was selected to preserve phytochemical compounds [23,24,25] for two days until its dry matter content reached at least 90%.

#### 2.1.4. Preparation of the Sugar Beet Fibrous Leaf Pulp (FLP) and Fibre-Rich Leaf Fraction (FRLF)

Fresh sugar beet leaves were used for the production of “fibrous leaf pulp” (FLP) and the “fibre-rich fraction” (FRLF). A volume of 240 mL of carbonate–bicarbonate buffer, pH 9.2 (Sigma-Aldrich, St. Louis, MO, USA), was added to the sugar beet leaves (1:8 (*w*/*v*)) and homogenised at 12,000 rpm for 3 min (Daihan HG-15D, Namyangju, South Korea). After homogenization, the mixture was extracted in a water bath (Nüve ST- 30, Ankara, Turkey) at 50 °C and 80 rpm for 30 min. At the end of extraction, coarse filtration was performed using cheesecloth, and the remaining cake was designated as “fibrous leaf pulp” (FLP). After coarse filtration, the liquid fraction was centrifuged (Nüve NF 800R, Ankara, Turkey) at 6000 rpm for 20 min and the remaining precipitate represented the “fibre-rich leaf fraction” (FRLF). For better understanding, a flow chart for the production of the “fibrous leave pulp” (FLP) and the “fibre-rich leaf fraction” (RLF) is shown in Figure 1.

### 2.2. Determination of Chemical Composition

#### 2.2.1. Determination of Moisture and Other Volatile Matter Content

In pre-dried metal containers, 3 g ± 0.001 g samples were weighed. The metal containers with the samples and the corresponding lids were placed in an electrically heated oven (Inko d.o.o., Zagreb, Croatia) set at 103 °C. The samples were dried for 4 h ± 0.1 h after the oven temperature had dropped back to 103 °C. The metal containers with the samples and the attached lids were removed from the oven and cooled to ambient temperature in the desiccator. The samples were weighed to the nearest 1 mg. The moisture content and volatile components were calculated from the difference in mass before and after drying according to the following equation [26]:(1)$$w=\frac{m_{0}-(m_{2}-m_{1})}{m_{0}}\times 100\%$$
where

m_0_ is the mass (g) of the test portion;

m_1_ is the mass (g) of the container with the lid;

m_2_ is the mass (g) of the container with the lid and test portion.

#### 2.2.2. Determination of Crude Ash

Combustion dishes containing 5 g ± 0.001 g of the test samples were placed on a hot plate and gradually heated until the test portions were carbonized. The combustion dishes were transferred to a muffle furnace (Inko d.o.o., Zagreb, Croatia) that had previously been heated to 550 °C, and left for 5 h. After cooling in the desiccator, the combustion dishes were weighed to the nearest 0.1 mg. The crude ash content was calculated from the difference in mass before and after combustion according to the following equation [27]:(2)$$w=\frac{m_{5}-m_{3}}{m_{4}-m_{3}}\times 100\%$$
where

m_3_ is the mass (g) of the empty dish;

m_4_ is the mass (g) of the dish containing the test portion;

m_5_ is the mass (g) of the dish and crude ash.

#### 2.2.3. Determination of Fat Content

Samples of 5 g ± 0.001 g were weighed into extraction tubes and covered with a fat-free wad of cotton wool. The thimbles were placed in Soxhlet extractors (Inko d.o.o., Zagreb, Croatia) and extracted with petroleum ether (Fisher Scientific Inc., Hampton, NH, USA) for 6 h. After removing the extraction flask, the solvent was distilled off. The residue in the bottles was dried in a drying oven (Inko d.o.o., Zagreb, Croatia) at 103 °C for 10 ± 0.1 min. After cooling in the desiccator, flasks were weighed to the nearest 0.1 mg. The fat content of the samples was determined by determining the weight of fat extracted according to the following equation [28]:(3)$$w=\frac{\left(m_{8}-m_{7}\right)}{m_{6}}\times f$$
where

m_6_ is the mass (g) of the test portion;

m_7_ is the mass (g) of the flask with silicon carbide chips;

m_8_ is the mass (g) of the flask with silicon carbide chips and dried light petroleum extract residue obtained;

f is the unit correction factor (f = 1000 g/kg).

#### 2.2.4. Determination of Nitrogen Content and Calculation of Crude Protein Content

The test portion of the samples 1 g ± 0.001 g were digested using a block digestion (FOSS, Hillerød, Denmark). Concentrated sulfuric acid (Sigma Aldrich, St. Louis, MO, USA) was used to convert the protein nitrogen to ammonium sulfate, at a boiling point elevated by the addition of potassium sulfate (Sigma Aldrich, St. Louis, MO, USA). A copper catalyst (Sigma Aldrich, St. Louis, MO, USA) was used to increase the reaction rate. An excess of sodium hydroxide (Sigma Aldrich, St. Louis, MO, USA) was added to the cooled digestion to release ammonia. The released ammonia was distilled using a semi-automatic steam distillation unit Foss Kjeltec 8200 (FOSS, Hillerød, Denmark). After distillation of the ammonia into an excess of boric acid solution (Sigma Aldrich, St. Louis, MO, USA), titration was performed with hydrochloric acid solution (Sigma Aldrich, St. Louis, MO, USA) to a colorimetric endpoint. The nitrogen content (w_N_) was calculated from the amount of ammonia produced according to the following equation:(4)$$w_{N}=\frac{1.4007(V_{S}-V_{b})\times c_{S}}{m_{9}}$$
where

V_S_ is the volume (mL) expressed to the nearest 0.05 mL of the HCl standard volumetric solution used in determination;

V_b_ is the volume (mL) expressed to the nearest 0.05 mL of the HCl standard volumetric solution used in the blank test;

c_S_ is the exact concentration (mol/L) expressed to four decimal places of the HCl standard volumetric solution used in the blank test;

m_9_ is the mass (g) of the test portion.

The crude protein content was determined by multiplying the result by the usual conversion factor of 6.25 [29].

#### 2.2.5. Determination of Crude Fibers

The test portion of the samples 1 ± 0.001 g were treated with boiling dilute sulfuric acid (Sigma Aldrich, St. Louis, MO, USA) for (30 ± 1) min. The residue was separated by filtration, washed and then treated with boiling potassium hydroxide solution (Sigma Aldrich, St. Louis, MO, USA) for 30 ± 1 min. The residue was separated by filtration and washed. The filter crucible was placed in a combustion dish and dried with its contents in a drying oven (Inko d.o.o., Zagreb, Croatia) at a temperature of 130 °C for 2 h. The filter crucible and the combustion bowl were cooled in a desiccator. Immediately after removal from the desiccator, the filter crucibles were weighed to the nearest 0.1 mg. The combustion dishes with the filter crucibles were placed on a hot plate and gradually heated until the test portions were carbonized. The combustion dishes were transferred to a muffle furnace (Inko d.o.o., Zagreb, Croatia), which had previously been heated to 500 °C, and left for 5 h. After cooling in the desiccator, the combustion dishes were weighed to the nearest 0.1 mg. The crude fiber content was calculated from the difference in mass before and after combustion according to the following equation [30]:(5)$$w=\frac{\left(m_{11}-m_{12}\right)}{m_{10}}$$
where

m_10_ is the mass (g) of the test portion;

m_11_ is the mass (g) of the combustion dish with the filter crucible containing the residue obtained after drying at 130 °C;

m_12_ is the mass (g) of the combustion dish with the filter crucible containing the residue obtained after ashing at 500 °C.

#### 2.2.6. Determination of Carbohydrates

Total carbohydrates were calculated according to the following equation:Total carbohydrates (%) = 100% − moisture (%) − ash (%) − fat (%) − protein (%) − fiber (%)(6)

#### 2.2.7. Determination of Sugars

The test portion of the samples 5 ± 0.001 g was mixed with 30 mL distilled water in an ultrasonic bath. The samples were transferred to volumetric flasks and mixed with 30 mL methanol (Sigma Aldrich, St. Louis, MO, USA). The flasks were filled up to 100 mL with distilled water. The extracts were filtered to remove particles and other interfering substances that could affect the chromatographic analysis. Sugars were determined using the high-performance liquid chromatography with refractive index detection (HPLC-RID) method. Chromatographic separations were performed on a Phenomenex Luna Omega 3 µm SUGAR 100A (LC column 150 × 3.0 mm) column with isocratic elution using acetonitrile:water (75:25, *v*/*v*). The column temperature was set to 30 °C and the flow rate to 1 mL/min.

The sugar content was calculated according to the following equation (Internal Method RU-MET-093, Edition 1):(7)$$w=\frac{A_{uz}}{A_{std}}\;\times \;c_{std}\times \;F_{r}$$
where

A_uz_ is the peak area of individual sugar in the sample;

A_std_ is the peak area of individual sugar in the standard solution;

c_std_ is the concentration (g/L) of the standard solution;

F_r_ is the dilution factor.

### 2.3. Determination of Water Activity (aw)

The water activity (aw) at 22.3 °C was determined using the aw meter Lab Swift-aw (Novasina AG, Lachen, Switzerland).

### 2.4. Thermal Analysis

Differential scanning calorimetry (DSC) was performed on dried powder samples of SBL, SBPE, SBPU, FLP and FRLF. The DSC Q1000 Differential Scanning Calorimeter (TA Instruments, New Castle, DL, USA) with an RCS cooling system was used. The instrument was calibrated for temperature and enthalpy according to the manufacturer’s standard instructions using the standard metal indium whose melting point temperature is T = 156.59 °C, and enthalpy of melting ΔH = 28.18 J/g [31]. All the experiments were conducted in nitrogen flow (purity stream of 99.999%) with a gas flow rate of 50 mL/min in a DSC cell, and 60 mL/min for thermogravimetric analysis (TGA). The SBL, SBPE, SBPU, FLP, FRLF samples for thermal analysis were weighed to 10–13 mg for TGA, and to 2–5 mg for DSC analysis using the analytical scale balance Mettler Toledo AE 163, (Mettler-Toledo Columbus, OH, USA). The samples for thermal analysis were taken after the homogenization of powders by mixing and the samples were taken as such for thermal analysis. All TGA and DSC scans were performed in triplicate. Samples for DSC analysis were crimped into aluminium pans by Blue press, TA and scanned in two cycles. The first cycle ranged from 20 to −90 °C, with a heating rate of 5 °C/min. After equilibration to −90 °C, the samples were scanned in the second cycle from −90 to 250 °C at a heating rate of 5 °C/min.

Thermogravimetric analysis (TGA) of SBL, SBPE, SBPU, FLP and FRLF samples was performed by TA Instruments TGA Q500 Thermogravimetric Analyser, (TA Instruments, New Castle, DL, USA) in nitrogen flow. The samples were scanned from room temperature to 700 °C, with a heating rate (Hr) of 5, 10 15 and 20 °C/min to obtain the kinetic parameters of the thermal decomposition of the samples. All scans were performed in triplicate.

### 2.5. Mathematical Models and Statistical Analysis

#### 2.5.1. Kinetics of Thermal Degradation

For the analysis of the kinetic parameters of the thermal degradation processes of SBL, SBPE, SBPU, FLP and FRLF samples, the peak corresponding to the thermal degradation of the samples in the temperature range of approximately 120 to 400 °C was utilized from the TGA curves. For this purpose, the samples were subjected to four heating rates: 5, 10, 15, and 20 °C/min. The methodology used to perform these kinetic studies was in accordance with the guidelines for data collection and computational analysis recommended by the ICTAC Kinetics Committee [32,33]. The activation energy (Ea) pertaining to the degradation process of sugar beet samples was determined using two non-isothermal isoconversional methods. The first method was the differential method formulated by Friedman [34] (Equation (8)), while the second method was the integral method devised by Ortega [35] (Equation (9)). The equations utilized were as follows:(8)$$\ln\left[\beta_{i}{\left(\frac{d\alpha }{dT}\right)}_{\alpha ,i}\right]=const-\frac{E_{\alpha }}{RT_{\alpha ,i}}$$(9)$$\ln\left(\frac{\beta_{i}}{\Delta T_{\alpha ,i}}\right)=const-\frac{E_{\alpha }}{RT_{\alpha ,i}}$$
where β represents the heating rate (K/min), α signifies the extent of conversion of the thermal degradation process, R denotes the universal gas constant (8.314 J/(molK)), and T stands for temperature (K). The subscripts i and α correspond to a specific heating rate and extent of conversion, respectively. In Equation (9), ΔT_α_ refers to T_α_ − T_α-Δα_, where Δα = 0.02. For each specific α, the E_α_ value is determined from the slope of a linear regression of the left side of Equations (8) and (9) plotted against 1/T_α,i_ (where T_α,i_ represents the temperature at which the extent of conversion α is achieved under the i-th heating rate). E_α_ values were computed within a range of α from 0.05 to 0.95 with an increment of 0.05 for both methods. The natural logarithm values of the pre-exponential factors at the extent of conversion α (ln(A_α_)) were determined utilizing the compensation effect, as described by Vyazovkin [36]. Subsequently, based on the Ea, A, and T values at the maximum degradation peak, the rate constants (k_p_) for the thermal denaturation process of sugar beet samples were calculated using the Arrhenius equation.

#### 2.5.2. Statistical Data Processing

All measurements were performed in triplicate and the results were expressed as means ± standard deviation (SD). XLSTAT (version 2014.5.03, Addinsoft, NY, USA), analysis and statistics add-in for MS Excel, was used for statistical analysis. The obtained values of the kinetic parameters Ea and lnA were subjected to a two-way ANOVA (factors: samples—five levels: SBL, FRL, FLP, SBPE and SBPU; models—two levels: Friedman and Ortega). To compare the means of these parameters, a post hoc Tukey’s test was used.

## 3. Results

### 3.1. Chemical Composition and Water Activity (aw)

The chemical composition and water activity of the SBL, FLP; FRLF, SBPU and SBPE samples are presented in Table 1.

The results presented in Table 1 indicate that there are significant differences in the values obtained for all chemical composition parameters and water activity (aw) for all samples according to ANOVA (*p* < 0.05). The protein content was the highest in the FRLF sample, followed by SBL and SBPE. The total fat content was the highest in FRLF, while the fat content in the other samples showed no significant variation. Carbohydrates were the most abundant in SBPU, followed by FLP and SBPE. The FRLF sample contains the highest protein content, while the FLP sample has the greatest carbohydrate content compared to the samples obtained from sugar beet leaves. This indicates that during the preparation process (extraction in a pH 9.2 buffer), each specific fraction is selectively enriched, with the FRLF being protein-rich and the FLP being carbohydrate-rich. The most protein-rich sample, the FRLF, is also the most fat-rich sample, suggesting that proteins and fats were co-separated during preparation of the FRLF.

All obtained results of the chemical composition (Table 1) are comparable to the results of thermogravimetric analysis, where the mass losses corresponding to moisture (first loss) and the mass loss related to the degradation of total organic matter, i.e., in our case, carbohydrates, proteins and fats (second and third loss), around 70%, are comparable with the results obtained by chemical composition analyses (see Table 2 for mass losses obtained by TGA).

For the FRLF, FLP, SBL and SBPE samples, the values obtained for moisture from TGA curves, presented in (Table 1) as the first loss, are 6.8 ± 1.1%, 7.2 ± 0.50%, and 4.8 ± 1.0%, while those obtained using standard moisture determination methods are 6.75 ± 0.01%, 7.5 ± 0.14%, 9.55 ± 0.07, and 5.75 ± 0.07%, respectively. Furthermore, the total organic matter values obtained for FRLF, FLP and SBL from TGA curves corresponding to losses (summed second, third, and fourth loss) are 61.13, 66.90, and 58.50, while those obtained using standard analytical methods are 76.83, 83.40 and 67.47, respectively.

Also, it can be seen that the FRLF sample contains the highest protein content, while the FLP sample has the greatest carbohydrate content compared to the samples obtained from sugar beet leaves. This indicates that during the preparation process (extraction in a pH 9.2 buffer), each specific fraction is selectively enriched, with the FRLF being protein-rich and the FLP being carbohydrate-rich. The most protein-rich sample, the fiber-rich leaf fraction (FRLF), is also the most fat-rich sample, suggesting that proteins and fats were co-separated during the preparation of the sample FRLF.

The values of water activity at a temperature of 22.3 °C obtained for SBLFLP, FRLF, SBPL and SBPU are presented in Table 1. All samples had aw values in the range from 0.288 to 0.428. The lowest value was found for SBL and the highest was found for FRLF. Although it has the highest moisture content (9.55%), the SBL sample had the lowest aw value, indicating that this sample can act as a humectant.

### 3.2. Thermal Analysis

#### 3.2.1. DSC Analysis

During the DSC experiments, phase changes but also chemical changes can occur in the process of temperature rising. In Figure 2, the DSC curves obtained for SBL, SBPE, SBPU, FLP and FRLF samples are presented.

In general, sugar beet samples had similar thermal behaviour, which is associated with a similar composition (Table 1). From the obtained DSC curves (Figure 2), it can be seen that in the temperature range from −90 to 0 °C, there were no thermal events observed, while in the temperature range from 0 to 250 °C, two endothermic transitions are visible accompanied by an exothermic heat flow deflection at around 200 °C: the first one in the temperature range from 31.2 to 134.4 °C for fibrous leaf pulp (FLP), from 38.9 to 160 °C for the sample of the fibre-rich leaf fraction (FRLF) and 34.3 to 136.9 °C for the sample of the sugar beet leaves (SBLs). The second transition was obtained in temperature range from 141.2 to 174.0 °C for the FLP sample, 160.3 to 212.9 °C for the sample of FRLF, and 153.2 to 182.0 °C for the sample of SBL. Similar results were obtained for sugar beet pellets (SBPE) and sugar beet pulp (SBPU) (Figure 2).

#### 3.2.2. TGA

Mass losses obtained from TGA/dTG curves for samples of fibrous leaf pulp (FLP), fibre-rich leaf fraction (FRLF), sugar beet leaves (SBLs), sugar beet pellets (SBPE) and sugar beet pulp (SBPU) obtained at a heating rate 5 °C/min are presented in Table 2.

In Figure 3, typical TGA (A) /dTG (B) curves, obtained for the dried, sugar beet leaves (SBLs), fibre-rich leaf fraction (FRLF), fibrous leaf pulp (FLP), sugar beet pellets (SBPE), and sugar beet pulp (SBPU) at a heating rate of 5 °C/min are shown.

All TGA curves obtained for different heating rates have similar thermal behavior with three main weight losses in the temperature range from 25 to 700 °C.

#### 3.2.3. Kinetic Analysis

The kinetic parameters of thermal degradation of the SBL, FRLF, FLP, SBPE, and SBPU were obtained using non-isothermal kinetic models. The dependence of activation energy on the extent of conversion (α) when applying the Friedman and Ortega isoconversion methods [34,35] was obtained.

Data for the calculation of the kinetic parameters were obtained from a series of TGA experiments performed at different heating rates, but with the same extent of conversion. The dependence of activation energy on the conversion rate for thermal degradation that took place in the temperature range from 120 to 400 °C is shown in Figure 3.

In Figure 4, the dependence of the activation energy Ea (A) and lnA (B) on the conversion (α) rate is presented. It can be concluded that there is a high similarity between the results obtained for the two models used. The activation energy (Ea) values obtained using both the Ortega and Friedman models for samples of sugar beet pulp (SBPU), sugar beet leaf pellets (SBPE), and sugar beet leaves (SBLs) ranged from 120 to 200 kJ/mol and ln A values for the dependence of the conversion rate (α) ranged from 25 to 50. The obtained Ea values for the fiber-rich leaf fraction (FRLF) and sugar beet fibrous leaf pulp (FLP) were in the range of 48 to 100 kJ/mol, while for the preexponential factor ln A, values were found to range from 5 to 20 for both Ortega and Friedman models (Figure 4A,B).

Based on the Ea, A, and T values at the maximum degradation peak, the rate constants (k_p_) for the thermal degradation process of sugar beet leaves samples were calculated using the Arrhenius equation. The obtained values for the kinetic triplet (Ea, ln A and k_p_) of the thermal decomposition of the samples FRLF, FLP, SBL, SBPE, and SBPU obtained using the differential (Friedman) and integral (Ortega) methods are presented in Table 3.

The kinetic triplets (Ea, lnA, and k_p_) of the thermal degradation process of the samples FRLF, FLP, SBL, SBPE, and SBPU were obtained using the differential Friedman and integral Ortega methods and are listed in Table 3. Based on the results of the two-way ANOVA, it can be concluded that there was no statistically significant difference between the applied models (*p* > 0.05). Sugar beet leaves (SBLs) exhibited the highest value of activation energy Ea, while sugar beet fibrous leaf pulp (FLP) had the lowest (*p* < 0.05).

Regarding the rate constants calculated for the temperature of the maximum value (peak) of the dTG curve, it can be observed that the constant rate of thermal degradation is the highest for the sugar beet leaf (SBL) sample concerning the kp_1_, and kp_2,_ which was obtained for the second maximum of the thermal degradation process, as degradation processes for leaves were exhibited in two stages, meaning that two weight losses on the TGA/dTG curve were detected. However, for samples of sugar beat FRLF and FLP, one instance of weight loss in the observed temperature range (from 150 to 400 °C) was found. It can be proposed that those differences were mainly a consequence of the different compositions of the samples.

## 4. Discussion

### 4.1. Chemical Composition and Water Activity

Sugar beet leaves contain 2.93% fiber, 21.2% hemicellulose, and 11.4% cellulose [15], whereas sugarcane leaves consist of 44% cellulose, 22% hemicellulose, and 17% lignin [37]. The results obtained for the chemical composition (Table 1) agree with those reported in the literature. Most of the structural carbohydrates are cellulose, hemicellulose and pectin, while lignin is present in small amounts [13]. The obtained value for carbohydrates of the SBP sample was found to be 78% in the present study, and it is known that sugar beet pulp contains 60–70% carbohydrates in dry matter, of which almost 90% is composed of cellulose, hemicellulose and pectin [8,13]. In the present study, the content obtained for proteins and total fat in the SBPU and SBL samples is comparable to the results reported by Zieminski et al. [38].

The obtained aw values are suitable considering microbial growth (aw ranged from 0.288 to 0.427) with the lowest value obtained for the samples of SBL and SBPE (aw = 0.288 and 0.226, respectively) and all samples can be considered as “low-moisture foods” (with water activity, aw < 0.6) [39].

### 4.2. Thermal Analysis

#### 4.2.1. DSC Analysis

Three thermal events in low-moisture biopolymer systems are directly linked to water–biopolymer interactions: the fusion of the frozen ‘‘free’’ water, the evaporation of the adsorbed water and the relaxation phenomenon. Relaxation is due to the stabilizing effect of water molecules on polysaccharides chains [40]. In the low humidity region, hydrogen polymer–polymer bonds stiffen the structure, and their breakage involves an endothermic peak of relaxation [40]. This relaxation endoderm, which is also connected to water evaporation, was also found for all samples (SBL, FRLF, FLP, SBPE, and SBPU) in the present work in the temperature range from about 30 °C to 150 °C, with a peak maximum at 76 °C, 81 °C, 98 °C 77 °C and 74 °C, respectively. Relaxation undergoes hysteresis [41], which has been observed for polysaccharides, and the temperature stays constant with the moisture content of the samples at around 50 °C. Rouilly et al. [40] found through the analysis of sugar beet pulp that the relaxation enthalpy increased to a maximum of 2.47 J/g of samples at a water content of 7.8%, which is also to the similar water content in the samples in the present study (Table 1 and Figure 2). The enthalpy of transition connected to water evaporation and relaxation obtained in the present study ranged from 150 to 170 J/g. This discrepancy compared to the value of 2.47 J/g obtained by Rouilly et al. [40] is a consequence of the different conditions of DSC experiments used in the present study (scanning in the non-hermetical Al pans) and can be expected. Rouilly et al. [40] found that the endothermal relaxation transition decreased until the peak complete disappearance for a water content exceeding 21.9%, and its temperature was constant around 52 °C. Considering the obtained onset temperatures (from 31 °C to 38 °C) of the endothermic transition, it is clear that these temperatures represent the limits beyond which the sample should not be treated or processed if its condition is to remain unaffected.

From the obtained DSC curves (Figure 2) of samples with the high carbohydrate content, it can be observed that SBPU and FLP had the high enthalpies for the first transitions (228.5 J/g and 170.7 J/g, respectively), which is also evident in the literature [40]. Despite having similar moisture percentages compared to the other samples, this finding suggests that the relaxation enthalpy contributed to the observed values, likely as a result of their higher carbohydrate content.

Huang at al. [42] observed an endothermic peak between 125 and 150 °C for the sugar beet pulp sample, as it was found in the present study in a similar temperature range (150 to 200 °C) for all samples (Figure 2). The peak was most likely due to thermal decomposition of the main components in the sugar beet samples, such as cellulose and hemicellulose [42].

The same authors [42] studied the influence of the particle size of the sugar beet pulp on thermal behaviour and concluded that particle size affected the DSC peak temperature, but no consistent trend was observed. This was observed in the present study, as the DSC transition peak temperatures slightly varied from sample to sample (Figure 2). It can be proposed that these variations were likely due to differences in particle size and the diverse chemical compositions of the samples. It has been concluded [42] that factors other than particle size might influence behaviour within this temperature range. They suggested that the exposure of polysaccharide and protein groups, which increases as particle size decreases, could reduce the melting temperature. However, they emphasized that further research is needed to clarify the specific mechanisms involved.

Sugar beet pulp and pellets are a complex mixture composed of 25% SBP pectin, 6–10% protein, and a total of 50% cellulose and hemicellulose [42]. In addition, sugar beet leaves have a complex composition and contain considerable amounts of protein (24–38%) [11,43], cellulose (13–18%), hemicellulose (11–17%), and pectin (14–18%) with small amounts of lignin (5–6%) [14].

#### 4.2.2. Thermogravimetry

Based on the TGA results (Figure 3 and Table 2), it can be suggested that the first mass loss observed for all samples, occurring in the temperature range of approximately 25 to 150 °C, generally corresponds to water evaporation. The second loss (from 150 to 400 °C) corresponds to the thermal degradation of mono- and disaccharides (fructose, glucose, sucrose) [44], and to the thermal degradation of proteins [45]. The third weight loss corresponds to the degradation of polymerized degradation products and thermal degradation of the cellulose and hemicellulose [46] in the sample [47]. Hemicellulose, cellulose, and lignin have been reported to decompose at temperatures in the ranges of 215–315 °C, 315–450 °C, and 200–900 °C, respectively [22,46,48]. The obtained percentages of mass loss do not directly reflect the composition of the samples, as during the thermal decomposition of complex samples, numerous components interact during the simultaneous increase in temperature, but some rough assumptions can be proposed [49]. Guo et al. [22] anticipated that slow thermal degradation in a N_2_ atmosphere could be a potential method for characterization and even identification of organic matter. Generally, the obtained results of thermogravimetric analysis in the present study can be roughly compared to the chemical composition (Table 1 and Table 2). The mass losses corresponding to moisture (first loss) and the mass loss related to the degradation of total organic matter, i.e., in our case, carbohydrates, proteins and fats (second and third loss sum equal around 70%), are comparable with the results obtained by chemical composition analyses (Table 1 and Table 2). Similar results were reported by Guo et al. [22], who characterized the organic matter of plants using thermal analysis in a nitrogen (N_2_) atmosphere.

Devrim [50] studied the pyrolysis kinetics of blends of lignite and sugar beet pulp and found the E_a_ value for sugar beet pulp to be about 97 kJ/mol. Uzun [48] found that the value of the activation energy for sugar beet pulp combustion was 14 kJ/mol. The differences found in the present study may be consequence of the sample composition varieties. It was found that the activation energy (Ea) value obtained for the thermal decomposition of SBL (Figure 3) is higher compared to the Ea value obtained for the thermal decomposition of all other samples, indicating that the sample of the sugar beet leaves (SBL) was thermally more stable compared to the other samples. The samples SBPE and SBPU follow SBL in their Ea values, indicating similar thermal stability, largely influenced by their chemical composition (Table 1). As is well known, carbohydrates can have a stabilizing effect on proteins, particularly in low-moisture samples [18]. It can be proposed that these stabilizing interactions influenced the thermal stability of the samples in the current study, enhancing stability in the sample with the optimal protein-to-carbohydrate ratio.

These findings align with the DSC results and the enthalpy values obtained for the second endothermic transition observed in these samples (Figure 2).

The activation energy (Ea) values for sugar beet leaf (SBL) thermal degradation and also for sugar beet leaf fractions (FRLF, SBPE and FLP) were presented for the first time in this work.

Kumar et al. [37] studied the pyrolysis of sugarcane leaves (*Saccharum officinarum* L.) using thermogravimetric analysis (TGA). The TGA results were analyzed using iso-conversional model-free methods as well as the multiple linear regression method. For a conversion rate (α) ranging from 0.05 to 0.95, the average apparent activation energy values derived from the iso-conversional methods were found to range from 214.9 to 239.6 kJ/mol. In the present study, the activation energy (Ea) values obtained for SBL and its fractions (FRLF and FLP) at a conversion rate (α) of 0.5–0.9 ranged between 50 and 200 kJ/mol. These values are comparable to those reported by Kumar et al. [37]. The observed differences, however, are likely due to variations in the chemical composition of the sugarcane leaves and sugar beet leaves.

Sugar beet leaves contain 2.93% fiber, 21.2% hemicellulose, and 11.4% cellulose [15], whereas sugarcane leaves consist of 44% cellulose, 22% hemicellulose, and 17% lignin [37]. The differences in fiber, hemicellulose, cellulose, and lignin content influence the thermal decomposition behavior of these biomass materials, leading to variations in Ea activation energy values.

Comparable results for activation energy (Ea) have also been reported for waste tobacco leaves [51]. Thermogravimetric analysis was conducted on waste tobacco leaves and stems under both air and inert (N_2_) atmospheres. The calculated average activation energy for tobacco leaves was 173.11–173.91 kJ/mol in air and 151.32–151.52 kJ/mol in nitrogen, while for tobacco stems, it was 183.28–185.63 kJ/mol in air and 168.50–168.81 kJ/mol in nitrogen [51]. It can also be proposed that slight variations in the Ea were the consequence of chemical composition of the leaf samples.

The Ea value for the sample of sugar beet leaves slightly increased with increases in the extent of conversion (α) (Figure 3), indicating reversible processes occurring not far from equilibrium [52]. The activation energy Ea for sugar beet pulp FLP and FRLF increased until the extent of conversion value reached 0.4, and with an increase in the α, Ea slightly decreased until the conversion reached the value of 1 for the FRLF sample, suggesting the complexity of the ongoing process, as a decreasing dependence is normally followed by a plateau associated with the activation energy of the forward process [52]. Ea values for samples of the sugar beet pulp SBPU and fibrous leaf pulp FLP continued to slightly decrease until the extent of conversion reached the value of 0.7 and then increased, indicating reversible processes [52].

Based on the Ea, A, and T values at the maximum degradation peak, the rate constants (k_p_) for the thermal degradation process of sugar beet leaf samples were calculated using the Arrhenius equation. As shown in Table 3, values for the kinetic triplet (Ea, ln A and k_p_) of the thermal decomposition of the sugar, FRLF, FLP, SBL, SBPE, and SBPU obtained using the differential (Friedman) and integral (Ortega) methods are in agreement with the results found for the thermal decomposition of the *Ginkgo biloba* leaves [53] and comparable with the results obtained for *Lippia origanoides* leaves [54].

## 5. Conclusions

A comprehensive thermoanalytical and kinetic study was conducted on by-products derived from the sugar beet industry. The samples analyzed included sugar beet waste leaves (SBLs), fractions obtained from sugar beet leaves (fiber-rich leaf fraction, FRLF, and fibrous leaf pulp, FLP), and sugar beet pulp (SBPU). During preparation, the fractions of sugar beet leaves (FRLF and FLP) were enriched with proteins and carbohydrates, respectively.

Differential scanning calorimetry (DSC) analysis revealed two endothermic transitions, accompanied by an exothermic heat flow deflection in all samples. The first endothermic transition was attributed to water evaporation and relaxation phenomena, consistent with the findings reported by Rouilly et al. [40]. In the present study, the enthalpy of relaxation and water evaporation ranged from 150 to 170 J/g, which differs from the value of 2.47 J/g reported by Rouilly et al. This discrepancy is likely due to differences in experimental conditions, such as the use of non-hermetical aluminum pans during scanning.

Samples with the highest carbohydrate content, SBPU and FLP, exhibited the highest enthalpy values for the first transition (213.9 J/g and 213.9 J/g, respectively). Despite having similar moisture percentages compared to the other samples, this finding suggests that the relaxation enthalpy contributed to the observed values, likely as a result of their higher carbohydrate content. The second endothermic peak, observed between 125 °C and 150 °C obtained for all samples, was due to the thermal decomposition of components such as cellulose and hemicellulose, which agree with findings reported by Huang et al. [42]. Variations in the peak temperatures of thermal transitions were attributed to differences in the chemical composition and particle size [42].

Thermogravimetric analysis (TGA) results (Figure 3, Table 2) showed three distinct mass loss stages. The first mass loss, occurring between 25 °C and 150 °C, corresponds to water evaporation. The second stage, observed between 150 °C and 400 °C, is associated with the thermal degradation of mono- and disaccharides (e.g., fructose, glucose, sucrose), as well as protein degradation. The third mass loss stage corresponds to the degradation of polymerized degradation products and the thermal decomposition of cellulose and hemicellulose [46]. Specifically, hemicellulose, cellulose, and lignin were reported to decompose in the temperature ranges of 215–315 °C, 315–450 °C, and 200–900 °C, respectively [22,46,48]. It is important to note that the obtained mass loss percentages do not directly reflect the chemical composition of the samples, as thermal decomposition involves complex interactions among components during temperature increases. However, rough assumptions can be made. The mass losses associated with moisture (first loss) and the degradation of total organic matter—primarily carbohydrates, proteins, and fats (second and third losses)—account for approximately 70% when taken together, which is in good agreement with the chemical composition results (Table 1 and Table 2).

The kinetic parameters for the thermal degradation of SBL, FRLF, FLP, SBPE, and SBPU were determined using a non-isothermal kinetic model. The dependence of activation energy (Ea) on the extent of conversion (α) was analyzed using the Friedman and Ortega iso-conversional methods.

In this study, the activation energies (Ea) for the thermal degradation of sugar beet leaves (SBLs) and their fractions (FRLF, SBPE, and FLP) were reported for the first time. Comparable Ea values have been reported for waste tobacco leaves [37,50,51]. The Ea for SBL thermal decomposition (Figure 3) was found to be higher than that of the other samples, indicating greater thermal stability. The SBPE and SBPU samples showed similar Ea values, suggesting similar thermal stability influenced by their chemical composition (Table 1). These results align with the DSC findings, particularly the enthalpy values for the second endothermic transition observed in these samples (Figure 2).

Using the activation energy (Ea), frequency factor (A), and peak degradation temperature (T), the rate constants (kp) for the thermal degradation of sugar beet leaf samples were calculated using the Arrhenius equation. The kinetic triplet (Ea, ln A, and kp_1_) values for the thermal decomposition of FRLF, FLP, SBL, SBPE, and SBPU obtained via the non-isothermal isoconversional methods are consistent with values reported for the thermal decomposition of *Ginkgo biloba* leaves [53] and comparable to results obtained for *Lippia origanoides* leaves [54].

## Figures and Tables

**Figure 1 foods-14-00307-f001:**
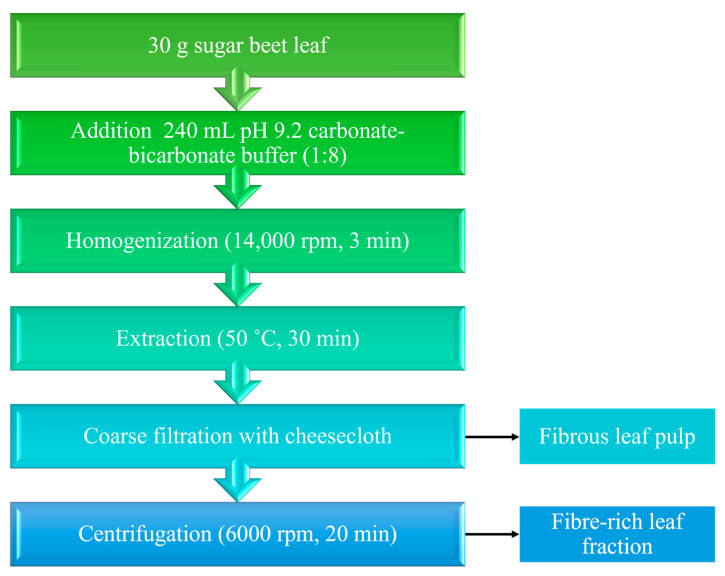
Flowchart of “fibrous leaf pulp” (FLP) and “fibre-rich leaf fraction” (FRLF) preparation.

**Figure 2 foods-14-00307-f002:**
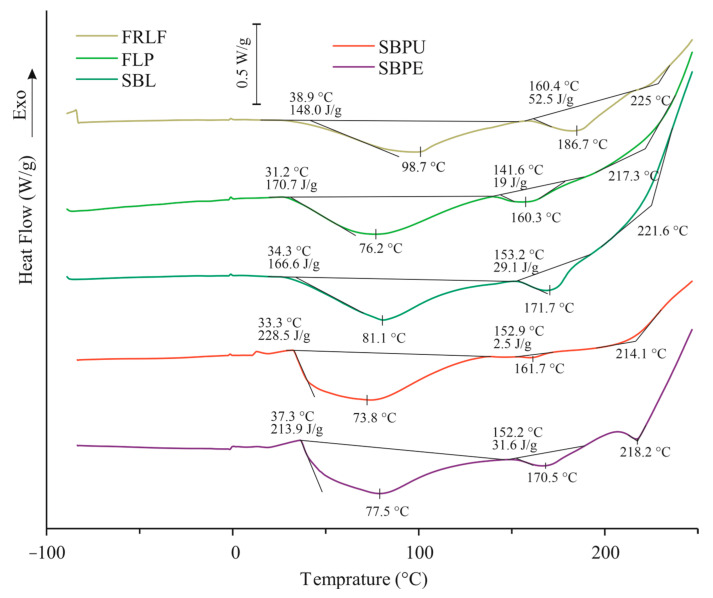
DSC curves for SBL, SBPE, SBPU, FLP and FRLF samples. The water evaporation, and the relaxation phenomenon (first peak), thermal decomposition (second peak) and exothermal deflection due to the total thermal degradation of the main components was detected for all samples.

**Figure 3 foods-14-00307-f003:**
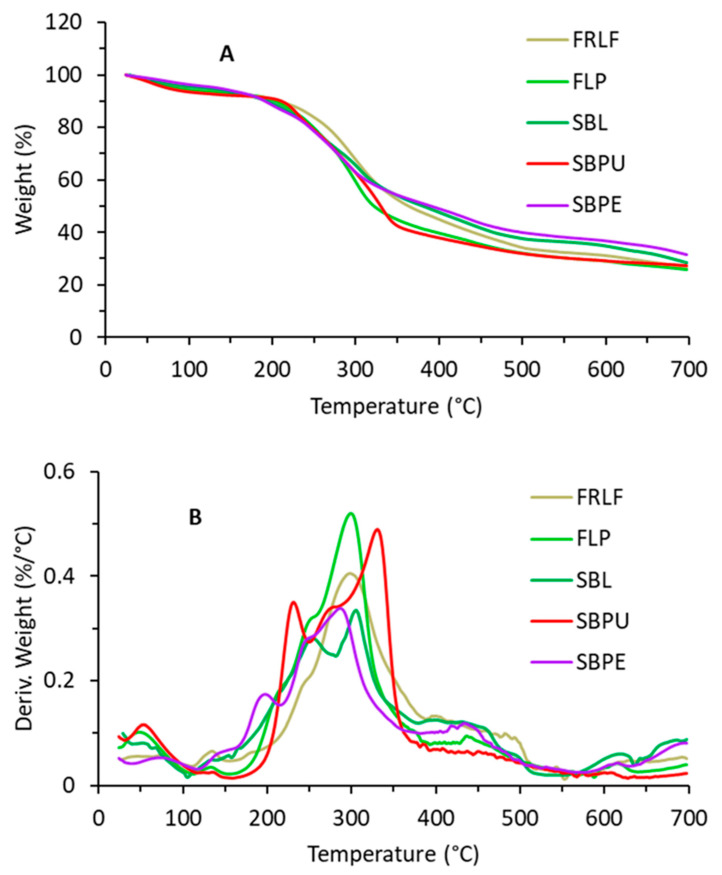
(**A**) TGA and (**B**) dTG curves of the dried SBL, FRLF, FLP, SBPE, and SBPU. The TGA and dTG curves exhibit three main weight loss stages, water evaporation (first loss) and thermal degradation (second and third losses).

**Figure 4 foods-14-00307-f004:**
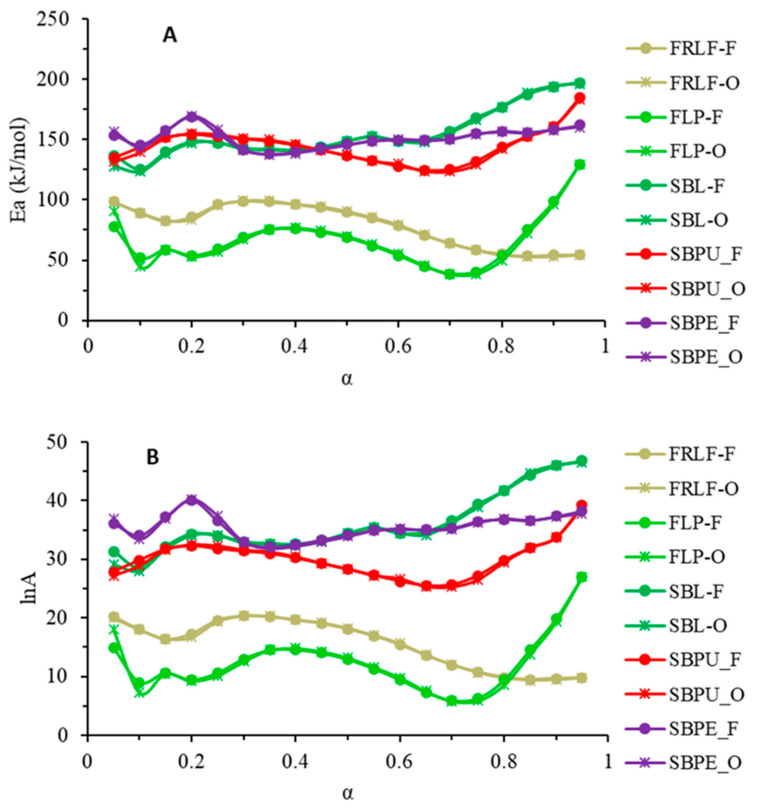
The dependence of (**A**) activation energy (Ea) and (**B**) lnA on the conversion rate (α) corresponding to the thermal decomposition of dried SBL, FRLF, FLP, SBPE, and SBPU; O: Ortega kinetic model; F: Friedman kinetic model.

**Table 1 foods-14-00307-t001:** Chemical composition and water activity (aw) of the sugar beet leaves SBL, FRL, FLP, SBPE and SBPU.

Parameter	Moisture (%)	aw	Proteins (%)	Total Fat (g/100 g)	Carbohydrates (%)	Ash (%)
SBL	9.55 ± 0.07 ^a^	0.288 ± 0.003 ^d^	20.52 ± 0.38 ^d^	2.76 ± 0.03 ^d^	44.17 ± 0.14 ^d^	23.00 ± 0.28 ^a^
FRLF	6.75 ± 0.01 ^d^	0.427 ± 0.008 ^a^	37.01 ± 0.05 ^a^	11.59 ± 0.20 ^a^	28.24 ± 0.36 ^e^	16.4 ± 0.14 ^c^
FLP	7.50 ± 0.14 ^b^	0.361 ± 0.010 ^b^	21.40 ± 0.13 ^c^	4.12 ± 0.18 ^b^	57.97 ± 0.09 ^b^	9.00 ± 0.00 ^d^
SBPE	5.75 ± 0.07 ^e^	0.226 ± 0.005 ^e^	27.63 ± 0.33 ^b^	3.46 ± 0.03 ^c^	45.65 ± 0.01 ^c^	17.50 ± 0.42 ^b^
SBPU	7.20 ± 0.00 ^c^	0.325 ± 0.003 ^c^	9.36 ± 0.14 ^e^	0.38 ± 0.02 ^e^	78.25 ± 0.30 ^a^	4.80 ± 0.14 ^e^

Data were subjected to one-way ANOVA (factor: samples—five levels: FRLF, SBL, FLP, SBPE, SBPU; degree of freedom was 4); different lowercase letters within the same column indicate a significant difference in means, according to Tukey’s HSD test (*p* < 0.05).

**Table 2 foods-14-00307-t002:** Mass losses obtained from TGA /DTG curves for samples of sugar beet obtained at heating rate 5 °C/min: fibrous leaf pulp (FLP), fibre-rich leaf fraction (FRLF), sugar beet leaves (SBLs), sugar beet pellets (SBPEs) and sugar beet pulp (SBPU).

	I Loss (%)	II Loss	III Loss (%)	IV Loss (%)	Total Loss (%)	Residue (%)
FLP	7.2 ± 0.5 ^a^	52.8 ± 0.6 ^a^	9.6 ± 0.5 ^c^	4.5 ± 0.8 ^d^	74.0 ± 1.3 ^a^	25.8 ± 1.3 ^b^
FRLF	6.8 ± 1.1 ^ab^	46.0 ± 0.9 ^b^	15.3 ± 0.6 ^b^	5.4 ± 0.7 ^cd^	73.3 ± 0.7 ^a^	26.7 ± 0.9 ^b^
SBL	4.8 ± 1.0 ^bc^	44.8 ± 0.9 ^b^	13.8 ± 0.9 ^b^	8.2 ± 1.2 ^c^	71.5 ± 1.3 ^a^	28.6 ± 1.3 ^ab^
SBPE	4.3 ± 0.8 ^c^	9.8 ± 1.0 ^d^	34.8 ± 0.8 ^a^	19.7 ± 1.2 ^b^	68.5 ± 0.8 ^b^	31.5 ± 0.9 ^a^
SBPU	7.2 ± 0.5 ^a^	13.9 ± 0.6 ^c^	11.3 ± 1.0 ^c^	40.5 ± 1.4 ^a^	72.8 ± 0.5 ^a^	27.1 ± 1.5 ^b^

Data were subjected to one-way ANOVA (factor: samples—five levels: FRLF, SBL, FLP, SBPE, SBPU; degree of freedom was 4); different lowercase letters within the same column indicate a significant difference in means, according to Tukey’s HSD test (*p* < 0.05).

**Table 3 foods-14-00307-t003:** The kinetic triplet (Ea, ln A and k_p_) of the thermal decomposition of the samples FRLF, FLP, SBL, SBPE, and SBPU obtained using the differential (Friedman) and integral (Ortega) methods.

		Ea (kJ/mol)	ln (A/min)	kp_1_ (1/min)	kp_2_ (1/min)
F	FRLF	79.1 ± 17.4 ^b^	15.6 ± 4.1 ^b^	0.43	-
	SBL	155.0 ± 20.1 ^a^	36.1 ± 5.1 ^a^	2.58	75.1
	FLP	66.3 ± 21.2 ^b^	12.4 ± 4.9 ^b^	0.19	-
	SBPE	151.2 ± 8.1 ^a^	35.5 ± 2.1 ^a^	0.082	39.6
	SBPU	144.3 ± 14.4 ^a^	30.0 ± 3.3 ^a^	0.024	5.2
O	FRLF	79.0 ± 17.6 ^b^	15.6 ± 4.2 ^b^	0.44	-
	SBL	154.1 ± 20.9 ^a^	35.8 ± 5.4 ^a^	2.55	70.4
	FLP	66.2 ± 22.3 ^b^	12.3 ± 5.2 ^b^	0.19	-
	SBPE	151.1 ± 8.5 ^a^	35.5 ± 2.2 ^a^	0.082	38.6
	SBPU	143.9 ± 14.7 ^a^	29.9 ± 3.4 ^a^	0.024	4.9

Data were subjected to two-way ANOVA (factors: samples—five levels: FRLF, SBL, FLP, SBPE, SBPU; degree of freedom was 4; models—two levels: Friedman and Ortega; degree of freedom was 1; interaction “models × samples”; degree of freedom was 4); different lowercase letters within the same column indicate a significant difference in means, according to Tukey’s HSD test (*p* < 0.05).

## Data Availability

D The original contributions presented in the study are included in the article, further inquiries can be directed to the corresponding author.

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
