# Peer review of "Thermal Characteristics and Kinetics of the Thermal Degradation of Sugar Beet Waste Leaves and Pulp in Relation to Chemical Composition"

_foods, 2025, doi:10.3390/foods14020307_

Round 1
Reviewer 1 Report
Comments and Suggestions for Authors
In this article, Thermal and kinetics characteristics of the sugar beet industry by-products and waste materials were investigated. These researches will promote the improvement of fully utilizing technology of the sugar beet industry by-products and waste materials. However, there are still some doubts in this work.
1、In Title and Abstract,why was this research done? Please polish the title and abstract.
2、In Introduce, from 31 to 45,I think that these introduces about Sugar beet cultivation are unnecessary, and this paragraph is redundant. Please polish this paragraph according to the needs of this article topic.
Comments on the Quality of English Language1、In line 78,“It is important to emphasize that sugar beet leaves as a valuable food additive, have....”should be“It is important to emphasize that sugar beet leaves, as a valuable food additive, have....”
2、In line133,the varieties of these plants used in this article should be introduced.
3、In line145 and 151,why was this condition,“at 55℃ for 48 hours.”,selected?Could it be a scope?
4、In line155 and 156,“„fibrous leaf pulp“” and“ 240 ml”are error.
5、From line182 and 213, “m0,m1...”should be accurately defined.
6、In line354 and 385,402 and 414, the abbreviation,“water activity (aw)”and“fibrous leaf pulp, (FLP)...” are repeated.
Author Response
Dear Sir/Madam,
The authors sincerely appreciate the valuable comments, corrections, and recommendations provided, which will undoubtedly enhance the quality of the manuscript. We are grateful for the time and effort the reviewers have dedicated to thoroughly evaluating our work.
We have made every effort to address all the reviewers' comments and suggestions and hope that the revisions meet their expectations.
Thank you for your consideration.
Faithfully,
Sanja Ostojic
Authors’ answers to the Reviewer 1:
Reviewer’s comment 1.
- In Title and Abstract, why was this research done? Please polish the title and abstract.
Authors’ answer: Authors are grateful for this comment. The title is changed and abstract was polished in accordance to revivers instructions.
New title Is as follows: Thermal characteristics and kinetics of the thermal degradation of waste sugar beet leaves and pulp in relation to chemical composition
In Abstract the environmental and industrial significance of our study was emphasized, what is the main cause for studying the thermal and kinetics characteristics of samples.
Lines 25-27: Results obtained are valuable for the sugar beet leaves industrial processing. Positive environmental impact is achieved by transforming the waste into high-value food additives
Reviewer’s comment 2.
- In Introduce,from 31 to 45,I think that these introduces about Sugar beet cultivation are unnecessary, and this paragraph is redundant. Please polish this paragraph according to the needs of this article topic.
Authors’ answer: Authors are thankful for this Reviewers’ comment. The text was polished and shortened. And now is as it follows:
Lines 33-42 Manuscript Revised: Sugar beet (Beta vulgaris L. vulgaris), is a biennial plant species of the Amaranthaceae family and is one of the four currently cultivated subspecies of beet and the only one used for sugar production [1]. Sugar beet cultivation is widespread worldwide, but generally thrives best in temperate climates between 30 and 60 ° north latitude [2]. In addition to sugar production, sugar beet is also important as a crop grown in rotation with other crops [3]. Today, sugar beet is the second most widely cultivated crop for sugar production, the first being, of course, sugar cane, which accounts for around 80% of total global sugar production [4]. Most sugar beet is grown in European countries, i.e. the European Union (EU) countries, Russia and Ukraine, and according to the European Commission (EC), EU countries account for around 50% of global production of this valuable sugar source.
Reviewer’s comment: Comments on the Quality of English Language
Reviewer’s comment 1:
1、In line 78 “It is important to emphasize that sugar beet leaves as a valuable food additive, have....”should be“It is important to emphasize that sugar beet leaves, as a valuable food additive, have....”
Authors’ answers: Authors are grateful for this reviewer’s observation; the coma was inserted and now the sentence is as follows:
Line 91 Manuscript Revised: It is important to emphasize that sugar beet leaves, as a valuable food additive, have significant development potential in dietary supplements due to their high-quality biological components.
Reviewer’s comment 2:
2、In line133, the varieties of these plants used in this article should be introduced.
Authors’ answers: Authors are grateful for this reviewer’s comment, the plant variety was added, as follows:
Line 148 Manuscript Revised: (Beta vulgaris L. vulgaris)
Reviewer’s comment 3:
3、In line145 and 151,why was this condition,“at 55℃ for 48 hours.”,selected?Could it be a scope?
Authors’ answers: Authors are grateful for this reviewer’s comment. The drying parameters for sugar beet leaves were determined as 55°C for 48 hours to prevent the degradation of bioactive compounds. This temperature and time combination is used to preserve the nutritional properties of plant materials, as excessive heat can negatively impact sensitive phytochemicals (López-Ortiz et al., 2024; ElGamal et al., 2023; Peñaloza et al., 2022).
Text is added as it follows:
Line 171-172 Manuscript Revised: selected to preserve phytochemical compounds (López-Ortiz et al., 2024; ElGamal et al., 2023; Peñaloza et al., 2022).
New References added to the manuscript:
- López-Ortiz, A., Salgado, M. N., Nair, P. K., Ortega, A. B., Méndez-Lagunas, L. L., Hernández-Díaz, W. N., & Guerrero, L. (2024). Improved preservation of the color and bioactive compounds in strawberry pulp dried under UV-Blue blocked solar radiation. Cleaner and Circular Bioeconomy, 9, 100112.
- ElGamal, R., Song, C., Rayan, A. M., Liu, C., Al-Rejaie, S., & ElMasry, G. (2023). Thermal degradation of bioactive compounds during drying process of horticultural and agronomic products: A comprehensive overview. Agronomy, 13(6), 1580.
- Peñaloza, S., Delesma, C., Muñiz, J., & López-Ortiz, A. (2022). The anthocyanin's role on the food metabolic pathways, color and drying processes: An experimental and theoretical approach. Food Bioscience, 47, 101700.
Reviewer’s comment 4:
4、In line155 and 156,“„fibrous leaf pulp“” and“ 240 ml”are error.
Authors’ answers: Authors are grateful for this reviewer’s comment and apologize for this error. The text is corrected and now is as follows:
Line 176 of the Manuscript Revised: The volume of 240 ml of carbonate-bicarbonate buffer, pH 9.
Reviewer’s comment 5:
5、From line182 and 213, “m0,m1...”should be accurately defined.
Authors’ answers: Authors are thankful for comment. To make it easier to understand, the numerical designations of the masses in the equations have been rearranged from the first to the last, so that the number next to the mass goes chronologically from 0 to 12 (Lines 202-280).
Reviewer’s comment 6:
6、In line354 and 385,402 and 414, the abbreviation,“water activity (aw)”and“fibrous leaf pulp, (FLP)...” are repeated.
Authors’ answers: Authors are grateful for this reviewer’s comment and apologize for this error. The text is corrected and repeated abbreviation were deleted (line 406 and line 426-429).
Reviewer 2 Report
Comments and Suggestions for Authors
The manuscript investigates the thermal and kinetic properties of sugar beet by-products, a relevant and innovative topic for both food industry sustainability and energy applications. The study utilizes differential scanning calorimetry (DSC) and thermogravimetric analysis (TGA) to examine these properties comprehensively. However, some sections require clarification, deeper discussion and improved presentation to strengthen the manuscript's impact.
Line 12-28: The abstract provides a good overview but lacks specific insights into the practical applications of the findings. Suggest including a sentence about the industrial or environmental implications of the study.
Line 46-50: The paragraph mentions the by-products of sugar beet processing but could elaborate on the environmental issues caused by these wastes to emphasize the significance of this research.
Line 75-77: The claim about the absence of studies on thermal characterization of dried sugar beet leaves should be supported by a reference or slightly rephrased for precision.
Line 137-145: The preparation steps for sugar beet leaves and pulp are detailed but lack clarity about replicates or quality control during sample preparation. This information would improve reproducibility.
Line 287-303: The DSC conditions should include why specific temperature ranges and rates were chosen. For instance, explain the rationale for scanning from -90°C to 250°C.
Line 345-369: Table 1 presents the chemical composition and water activity. Clarify whether the differences among samples are statistically significant for all parameters, and link these results to thermal behavior.
Line 393-405: The DSC analysis mentions "two endothermic transitions" but does not sufficiently explain their significance. Relating these transitions to practical processing parameters would improve the discussion.
Figures 2 and 3: The captions should provide more context, such as highlighting the significance of specific peaks or transitions visible in the curves.
Table 2 (Line 411): Ensure the presentation of data aligns with the discussion. Include explanations for why some samples exhibit higher thermal stability.
Line 571-573: The statement about sugar beet leaves being more thermally stable requires additional interpretation. Discuss how chemical composition (e.g., protein and carbohydrate content) influences this behavior.
Line 127-130: The sentence beginning with "The aim of the present study" is convoluted. Rephrase for clarity.
Line 319-329: Simplify the equations to make them more accessible to readers unfamiliar with the methodology.
Similarity index is too high to be published according to iThenticate report.
Author Response
Dear Sir/Madam,
The authors sincerely appreciate the valuable comments, corrections, and recommendations provided, which will undoubtedly enhance the quality of the manuscript. We are grateful for the time and effort the reviewers have dedicated to thoroughly evaluating our work.
We have made every effort to address all the reviewers' comments and suggestions and hope that the revisions meet their expectations.
Thank you for your consideration.
Faithfully,
Sanja Ostojic
Authors’ answers to the Reviewer 2:
Reviewer’s comment 1.
Line 12-28: The abstract provides a good overview but lacks specific insights into the practical applications of the findings. Suggest including a sentence about the industrial or environmental implications of the study.
Authors’ answer: Authors are thankful for useful reviewers’ comment. Processing waste sugar beet leaves into high-value food additives has significant industrial and environmental implications. In nowadays there are high interest for Value-Added Product Development especially if those products were obtained from waist lake is sugar beet leaves and pulp. There are several important benefits for turning the waste into highly valuable nutrition products and food additives. Converting waste sugar beet leaves and pulp into food additives creates high-value products such as dietary fibers, antioxidants, vitamins, and bioactive compounds. These additives can be used in functional foods, nutraceuticals, or as natural preservatives, broadening their market potential. In the same time there is positive environmental impact thru reducing the agricultural, biological waste (Aït-Kaddour et al 2024).This approach contributes to the broader goals of a circular economy and sustainable development(Ptak et al 2022, Aït-Kaddour et al 2024).
The following corrections have been made to the manuscript: the sentences have been added and highlighted in yellow, the Abstract has been adjusted accordingly to reflect this inclusion.
line 25-27: Results obtained are valuable for the sugar beet leaves industrial processing. Positive environmental impact is achieved by transforming the waste into high-value food additives.
References:
Ptak, M.; Skowronska, A.;Pinkowska, H.; Krzywonos, M. Sugar Beet Pulp in the Context of Developing the Concept of Circular Bioeconomy. Energies 2022, 15, 175. https://doi.org/10.3390/en15010175
Abderrahmane Aït-Kaddour, Oumayma Boukria, Abdo Hassoun , Yana Cahyana, Ines Tarchi , Fatih Ozogul, Mohammed Loudiyi, Khaoula Khwaldia; Transforming plant-based waste and by-products into valuable products using various “Food Industry 4.0” enabling technologies: A literature review , Science of the Total Environment 955 (2024) 176872
https://doi.org/10.1016/j.scitotenv.2024.176872
Reviewer’s comment 2.
Line 46-50: The paragraph mentions the by-products of sugar beet processing but could elaborate on the environmental issues caused by these wastes to emphasize the significance of this research.
Authors’ answer:
The authors are grateful to the Reviewer for highlighting the importance of considering the environmental impact of the present study. In response, the following text, highlighted in yellow, has been added to emphasize the environmental significance.
Text is added (lines 48-64) in the Manuscript as follows:
Lines 48-64: Reducing food waste not only benefits the environment but also contributes to various social agendas by reducing greenhouse gas emissions, enhancing ecosystem services, and enforcing sustainable resource management policies. Food waste accounts for 8-10 % of global greenhouse gas emissions and ranks as the third-largest emitter globally, following China and the United States (Aït-Kaddour et al 2024). In order to use resources efficiently and enhance food security and sustainability, food waste and by-products must be reduced and properly valorized. Plant-based food production generates various by-products which are generally rich in nutrients and bioactive compounds.
In order to achieve Sustainable Development Goals (SDGs), the food industry and others are trying to minimize the final amount of accumulated waste and by-products. This is being pursued through increased recycling efforts and repurposing these materials for alternative applications. In addition to the accumulation of a large amount of waste and by-products, sugar production also has a significant impact on the environmental factors. Namely, a life cycle assessment study found that the production of 1 ton of white consume sugar generates up to 1540 kg of CO2 equivalents and consumes up to 12.6 m3 of water (Namdari et al., 2022). Emerging technologies have been effectively employed to extract these valuable compounds with health benefits (Aït-Kaddour et al 2024).
Lines 139-142: Therefore, it is of utmost importance to know the kinetics and thermal characteristics so that future potential non-thermal processing methods for utilization of waste and by-products from the sugar industry can be theoretically optimized to ensure their reuse and reduce negative environmental impacts.
References added to the Manuscript revised:
Abderrahmane Aït-Kaddour, Oumayma Boukria, Abdo Hassoun , Yana Cahyana, Ines Tarchi , Fatih Ozogul, Mohammed Loudiyi, Khaoula Khwaldia; Transforming plant-based waste and by-products into valuable products using various “Food Industry 4.0” enabling technologies: A literature review , Science of the Total Environment 955 (2024) 176872
https://doi.org/10.1016/j.scitotenv.2024.176872
Namdari, M., Rafiee, S., Notarnicola, B., Tassielli, G., Renzulli, P. A., & Hosseinpour, S. (2022). Use of LCA indicators to assess Iranian sugar production systems: case study — Hamadan Province. Biomass Conversion and Biorefinery. https://doi.org/10.1007/S13399-022-02982-4
Reviewer’s comment 3.
Line 75-77: The claim about the absence of studies on thermal characterization of dried sugar beet leaves should be supported by a reference or slightly rephrased for precision.
Authors’ answer:
Authors are very thankful for that observation.
The sentence in the Manuscript Revised on lines 133-134 was rephrased for precision and changed as follows:
Lines 133-134: A literature search revealed no studies investigating the thermal characteristics of sugar beet industry waste, specifically sugar beet leaves and pulp.
Reviewer’s comment 4.
Line 137-145: The preparation steps for sugar beet leaves and pulp are detailed but lack clarity about replicates or quality control during sample preparation. This information would improve reproducibility.
Authors’ answer:
Authors are grateful for Reviewer’s observation. The preparation steps for sugar beet leaves and pulp are presented in cleared whey regarding replicates and quality control during sample preparation. In sample preparation process, all experiments were conducted in triplicates, and quality control was ensured by consistently monitoring sample weights, drying times, and temperatures. It has been added to the manuscript as follows:
The text in Revised Manuscript, lines 155 -157: All experiments were conducted in triplicates, and quality control was ensured by consistently monitoring sample weights, drying times, and temperatures.
Lines 163-164: All experiments were conducted in triplicates, and quality control was ensured by consistently monitoring sample weights, drying times, and temperatures.
Lines 171-172: selected to preserve phytochemical compounds (López-Ortiz et al., 2024; ElGamal et al., 2023; Peñaloza et al., 2022).
References added to the text of the Manuscript revised:
López-Ortiz, A., Salgado, M. N., Nair, P. K., Ortega, A. B., Méndez-Lagunas, L. L., Hernández-Díaz, W. N., & Guerrero, L. (2024). Improved preservation of the color and bioactive compounds in strawberry pulp dried under UV-Blue blocked solar radiation. Cleaner and Circular Bioeconomy, 9, 100112.
ElGamal, R., Song, C., Rayan, A. M., Liu, C., Al-Rejaie, S., & ElMasry, G. (2023). Thermal degradation of bi-oactive compounds during drying process of horticultural and agronomic products: A comprehensive over-view. Agronomy, 13(6), 1580.
Peñaloza, S., Delesma, C., Muñiz, J., & López-Ortiz, A. (2022). The anthocyanin's role on the food metabolic pathways, color and drying processes: An experimental and theoretical approach. Food Bioscience, 47, 101700
Reviewer’s comment 5.
Line 287-303: The DSC conditions should include why specific temperature ranges and rates were chosen. For instance, explain the rationale for scanning from -90°C to 250°C.
Authors’ answer:
Arthurs are grateful for reviewers’ question.
In order to thermally characterize the samples, it is desirable to follow the thermal changes caused by the change in temperature in the widest possible temperature range in which the sample can undergo various thermal events and thus show its characteristics. A specific temperature range, from -90 to 250 °C was applied to enable inducing thermal events that take place at low temperatures, characteristic for samples with a small amount of water and a low aw value. Samples in present study were composed of the different sugars, proteins and dietary fibers and as it is known that the glass transition occurs at 100–150 °C below the component melting temperature, which has been confirmed to apply to sugars (Roos 2003) the used temperature range was appropriate. At the temperature higher than 200 °C the thermal decomposition of the sample begun as it is showed on the DSC curves (Fig 2) as an exothermic heat flow deflection at around 200 °C. As DSC technique is used to define the thermal transitions caused by physicochemical events which can also define structural and conformational changes in sample, the higher temperatures (over 250°C) were not implied. The thermal decomposition was defined by another thermoanalytical technique Thermogravimetry, which is appropriate for those studies.
Sugars are well-known glass formers (Roos 2021) are present in samples of the sugar beet leaves, leaves fraction and pulp, and intention was to check over wide the temperature rang possibility of glass transition existence.
A glass transition is a property of a nonequilibrium material that occurs as a reversible transition between solid-like (glassy) and liquid-like (rubbery, leathery, syrupy) states of an amorphous (disordered) material. Foods in their frozen state contain ice crystals imbedded in a noncrystalline, amorphous, unfrozen fluid (syrup). This is typically because sugars or a mix of multiple solutes in the aqueous phase often show delayed or inhibited solute crystallization. At a low temperature, solutes in the absence of solute crystallization become highly freeze-concentrated, and such freeze-concentrated solutes with some unfrozen water vitrify without crystallization as temperature is sufficiently lowered (Roos 2021).
Pure food components, such as amorphous sugars, often show a single, clear glass transition that can be observed following dielectric, mechanical or thermal properties.
In many real food systems, components may be only partially amorphous (carbohydrates, e.g., starch and sugars; proteins) or many food components may only be partially miscible or immiscible at all with each other, forming single or several phases at microstructure scale (Roos, 2003). Recently several authors evidenced that composite materials may present the thermal signature of two glass transitions in differential scanning calorimetry, reflecting the possible inhomogeneity of the structure with different areas more or less rich in plasticizing molecules (Masavang et al 2019). For all these reasons the temperature range from -90°C to 250°C was applied in present study.
References:
- H. Roos, Thermal analysis, state transitions and food quality, Journal of Thermal Analysis and Calorimetry, Vol. 71 (2003) 197–203
Yrjö H Roos Glass Transition and Re-Crystallization Phenomena of Frozen Materials and Their Effect on Frozen Food Quality, Foods, 2021;10(2):447.
doi: 10.3390/foods10020447
Supuksorn Masavang, Gaëlle Roudaut, Dominique Champion, Identification of complex glass transition phenomena by DSC in expanded cereal-based food extrudates: Impact of plasticization by water and sucrose, Journal of Food Engineering, 245, 2019, 43-52.
https://doi.org/10.1016/j.jfoodeng.2018.10.008
Reviewer’s comment 6.
Line 345-369: Table 1 presents the chemical composition and water activity. Clarify whether the differences among samples are statistically significant for all parameters, and link these results to thermal behavior.
Authors’ answer:
Thank you for your observation. We appreciate the opportunity to clarify this aspect of the manuscript. The differences among the samples in Table 1 have been statistically evaluated using one-way ANOVA, with significant differences (p < 0.05) identified for all parameters, as indicated by the distinct lowercase letters within each column. This analysis confirms that the chemical composition and water activity vary significantly across the samples.
Although there are statistically significant differences in the chemical composition, this does not highly affect the thermal characteristics of the samples.
Sentence in the Manuscript was modified (lines 375-376):
Lines 375-376: The results presented in Table 1 indicate that there are significant differences in the values obtained for all chemical composition parameters and water activity (aw), for all samples according to ANOVA (p<0.05).
Reviewer’s comment 7.
Line 393-405: The DSC analysis mentions "two endothermic transitions" but does not sufficiently explain their significance. Relating these transitions to practical processing parameters would improve the discussion.
Authors’ answer:
Authors are thankful for this reviewer’s comment.
Two endothermic transitions observed in the DSC curves indicate the temperature limits beyond which the sample should not be treated or processed if its condition is to remain unaffected. The first limiting temperature, observed between 31 and 38 °C, which is the onset temperature of the endothermic transition beginning, clearly suggests that the sample should be processed at low temperatures (30 °C).
However, if the objective is to transform the sample into a state where the first endothermic transition is eliminated, the sample must be processed at approximately 150 °C. As noted in the manuscript (lines 522 to 525): “In the low humidity region, hydrogen polymer–polymer bonds stiffen the structure, and their breakage involves an endothermic peak of relaxation [38]. This relaxation endotherm, which is also associated with water evaporation, was observed for all samples..."
The first endothermic transition corresponds to the polysaccharide chain relaxation endotherm and water evaporation. Additionally, protein denaturation occurs within the same temperature range. Therefore, it can be assumed that all proteins present in the samples are in a denatured state after processing at temperatures around 150 °C (Ostojic et al., 2023).
The following sentence was added into Revised Manuscript -Discussion:
Lines 540-543: Considering the obtained onset temperatures (from 31°C to 38°C) of the endothermic transition, it is clear that these temperatures represent the limits beyond which the sample should not be treated or processed if its condition is to remain unaffected.
Reference:
Ostojić, S.; Micić, D.; Zlatanović, S.; Lončar, B.; Filipović, V.; Pezo, L. Thermal Characterisation and Isoconversional Kinetic Analysis of Osmotically Dried Pork Meat Proteins Longissimus Dorsi. Foods 2023, Vol. 12, Page 2867 2023, 12, 2867, doi:10.3390/FOODS12152867
Reviewer’s comment 8.
Figures 2 and 3: The captions should provide more context, such as highlighting the significance of specific peaks or transitions visible in the curves.
Authors’ answer: Authors are grateful for tis observation. Highlighting the significance of the specific peaks in Figures 2 and 3 is provided by adding the text in Figures 2 and 3 captions as follows:
Line 417-419: The water evaporation, and the relaxation phenomenon (first peak), thermal decomposition (second peak) and exothermal deflection due to the total thermal degradation of the main components was detected for all samples.
Lines 445-447: The TGA and dTG curves exhibit three main weight loss stages, water evaporation (first loss) and thermal degradation (second and third losses).
Reviewer’s comment 9.
Table 2 (Line 411): Ensure the presentation of data aligns with the discussion. Include explanations for why some samples exhibit higher thermal stability.
Authors’ answer: Authors are thankful for Reviewers useful comment. The authors apologize for this error. The error was corrected and now the sample names are listed correctly.
Lines 431-433 Manuscript Revised the text was added as follows: Fibrous leaf pulp (FLP), Fibre rich leaf fraction (FRLF), Sugar beet leaves (SBL), Sugar beet pellets (SBPE), Sugar beet pulp (SBPU)
Line 468-473 of the Manuscript revised: The errors were corrected to ensure that the presentation of the data aligns with the discussion. The revised text is now as follows:
Lines 468-473: samples of sugar beet pulp (SBPU), sugar beet leaf pellets (SBPE), and sugar beet leaves (SBL) ranged from 120 to 200 kJ/mol and ln A values in the dependence of the conversion rate (α) ranged from 25 to 50. The obtained Ea values for the fiber-rich leaf fraction (FRLF) and sugar beet fibrous leaf pulp (FLP) were in the range of 48 to 100 kJ/mol, while for preexponential factor ln A values were found to range from 5 to 20, for booth Ortega and Friedman models (Figure 4.A and B).
In the line 489 Manuscript Revised: the sample names were corrected, and now is as follows:
FRLF, FLP, SBL, SBPE, and SBPU,
Reviewer’s comment:
Include explanations for why some samples exhibit higher thermal stability.
Authors’ answer: this part of the response is incorporated in the following Authors’ answer, because of similarity of the reviewer’s request. The main reason why some samples exhibit higher thermal stability, as stated in the manuscript, is attributed to differences in their chemical composition.
Reviewer’s comment 10.
Line 571-573: The statement about sugar beet leaves being more thermally stable requires additional interpretation. Discuss how chemical composition (e.g., protein and carbohydrate content) influences this behavior.
Authors’ answer: Authors are thankful for Reviewers observation.
The sample of sugar beet leaves (SBL) can be considered more thermally stable and less prone to thermal degradation compared to the other samples. This conclusion is supported by the activation energy (Ea) values obtained for the SBL sample, as shown in Table 3. The Ea value for SBL is the highest compared to the other samples, indicating that more energy is required to initiate the thermal decomposition process. Therefore, it can be concluded that the sugar beet leaves sample exhibits the highest thermal stability among the analyzed samples.
As is well known, carbohydrates can have a stabilizing effect on proteins, particularly in low-moisture samples (Ostojic et al., 2023). It can be proposed that these stabilizing interactions influenced the thermal stability of the samples in the current study, enhancing stability in the sample with the optimal protein-to-carbohydrate ratio. Interestingly, the SBPU sample, with the highest carbohydrate content (79%) and the lowest protein content (9%), and the SBPE sample, with 46% carbohydrates and 27% protein, exhibit similar activation energy (Ea) values of 144.3 kJ/mol and 151.2 kJ/mol, respectively.
These interactions raise many new questions and require deeper investigation in the future. (Zhang et al 2021)
Line 600-603 Manuscript Revised following text is added: As is well known, carbohydrates can have a stabilizing effect on proteins, particularly in low-moisture samples (Ostojic et al., 2023). It can be proposed that these stabilizing interactions influenced the thermal stability of the samples in the current study, enhancing stability in the sample with the optimal protein-to-carbohydrate ratio.
References:
Ostojić, S.; Micić, D.; Zlatanović, S.; Lončar, B.; Filipović, V.; Pezo, L. Thermal Characterisation and Isoconversional Kinetic Analysis of Osmotically Dried Pork Meat Proteins Longissimus Dorsi. Foods 2023, Vol. 12, Page 2867 2023, 12, 2867, doi:10.3390/FOODS12152867.
Zhang S, Chen KY, Zou X. Carbohydrate-Protein Interactions: Advances and Challenges. Commun Inf Syst. 2021;21(1):147-163. doi: 10.4310/cis.2021.v21.n1.a7. PMID: 34366717; PMCID: PMC8336717.
Reviewer’s comment 11.
Line 127-130: The sentence beginning with "The aim of the present study" is convoluted. Rephrase for clarity.
Authors’ answer: Authors are thankful for Reviewers useful comment. The sentence was rephrased and added into Manuscript Revised.
Lines 142-144 Manuscript Revised: The aim of the present study was to perform a thermal characterization of dried sugar beet pulp, pellets, leaves, and various leaf fractions.
Reviewer’s comment 12.
Line 319-329: Simplify the equations to make them more accessible to readers unfamiliar with the methodology.
Authors’ answer:
Thank you for your valuable feedback. The equations presented in the manuscript are indeed in their simplified form and are widely recognized and accepted in the field of thermal analysis and kinetics of thermal processes. Specifically:
Equation (8) is the differential isoconversional method proposed by Friedman, and
Equation (9) is the integral isoconversional method developed by Ortega.
Both methodologies are standard tools in thermal kinetics studies, and references [31] and [32] are provided to guide readers to the original sources for further details.
Furthermore, the equations were intentionally presented with sufficient clarity and accompanied by definitions of all terms to ensure accessibility. While it is true that a certain level of familiarity with mathematical formulations is expected from the readership of this journal, we have aimed to strike a balance between providing essential methodological details and maintaining readability.
We believe that retaining these equations in their current form is critical for the scientific rigor and transparency of our work.
Reviewer’s comment 13
Similarity index is too high to be published according to iThenticate report.
Authors’ answer:
Authors are thankful for this Reviewer’s s comment.
The paper was checked using the Turnitin program. Authors excluded their names, institutions, references, and similar elements from the paper, which resulted in a decrease from 38% to 22%. In MDPI check, they did not exclude such elements, which is why the percentage of "similarity" is so high.

Round 2
Reviewer 2 Report
Comments and Suggestions for Authors
It can be publisheable from my side.